# The Mediation Effect of Pain on the Relationship between Kinesiophobia and Lumbar Joint Position Sense in Chronic Low Back Pain Individuals: A Cross-Sectional Study

**DOI:** 10.3390/ijerph20065193

**Published:** 2023-03-15

**Authors:** Mohammad A. ALMohiza, Ravi Shankar Reddy, Faisal Asiri, Adel Alshahrani, Jaya Shanker Tedla, Snehil Dixit, Kumar Gular, Venkata Nagaraj Kakaraparthi

**Affiliations:** 1Department of Rehabilitation Sciences, College of Applied Medical Sciences, King Saud University, Riyadh 11362, Saudi Arabia; 2Department of Medical Rehabilitation Sciences, College of Applied Medical Sciences, King Khalid University, Abha 61421, Saudi Arabia; 3Department of Physical Therapy, College of Applied Medical Sciences, Najran University, Najran 55461, Saudi Arabia

**Keywords:** low back pain, lumbar spine, kinesiophobia, position sense, pain severity

## Abstract

(1) Background: Fear of movement (kinesiophobia) and impaired lumbar joint position sense (LJPS) play a vital role in developing and maintaining non-specific chronic low back pain (CLBP). However, how kinesiophobia impacts LJPS is still being determined. The aims of this study are to (1) assess the correlation between kinesiophobia and LJPS in individuals with chronic low back pain; (2) compare LJPS between individuals with CLBP and those who are asymptomatic; and (3) evaluate if pain can mediate the relationship between kinesiophobia and LJPS in CLBP individuals. (2) Methods: Eighty-three individuals (mean age = 48.9 ± 7.5 years) with a diagnosis of CLBP and 95 asymptomatic individuals (mean age = 49.4 ± 7.0 years) were recruited into this cross-sectional study. Fear of movement in CLBP individuals was assessed using the Tampa Scale for Kinesiophobia (TSK). LJPS was determined using the active target repositioning technique using a dual-digital inclinometer. LJPS was evaluated in lumbar flexion, extension, and side-bending left and right directions, and the repositioning accuracy was determined in degrees using a dual digital inclinometer. (3) Results: Kinesiophobia showed a significant (*p* < 0.001) moderate positive correlation with LJPS (flexion: r = 0.51, extension: r = 0.41, side-bending left: r = 0.37 and side-bending right: r = 0.34). LJPS errors were larger in CLBP individuals compared to asymptomatic individuals (*p* < 0.05). Mediation analyses showed that pain significantly mediated the relationship between kinesiophobia and LJPS (*p* < 0.05) in CLBP individuals. (4) Conclusions: Kinesiophobia and LJPS were positively associated. LJPS is impaired in CLBP individuals compared to asymptomatic individuals. Pain may mediate adverse effects on LJPS. These factors must be taken into account when assessing and developing treatment plans for those with CLBP.

## 1. Introduction

Low back pain is a highly prevalent and significant musculoskeletal disorder affecting the general population, leading to high healthcare costs [1,2]. Low back pain affects up to 20% of the general population, and one-year prevalence is 36 to 38% [3]. In addition, the duration of symptoms is directly tied to the condition’s prognosis; the longer low back pain persists, the worse the prognosis [4]. Individuals with chronic low back pain (CLBP) experience pain, functional disability, fear, and anxiety, which can significantly impact their quality of life [5,6].

Fear of movement or activity (kinesiophobia) is a psychological factor associated with the severity and persistence of pain, which has garnered much attention in individuals with chronic pain [7,8]. Kinesiophobia is a term that describes excessive fear of performing a bodily movement due to the expectation of injury or re-injury [9,10]. The prevalence of kinesiophobia is over 50% in chronic pain individuals [11]. The precise way of assessing kinesiophobia is to measure with the Tampa Scale for Kinesiophobia (TSK) [12]. Higher TSK scores are associated with increased pain and disability [13,14].

The etiology of CLBP is complex and multifactorial, making it challenging to identify a single specific element that contributes to and maintains chronic pain [15]. Joint position sense is a critical factor that significantly contributes to the stability of the lumbar spine [16]. Decreases in muscle strength, endurance, and impaired lumbar joint position sense (LJPS) are documented in CLBP individuals, and these factors may significantly maintain the symptoms for a longer duration [17,18]. Additionally, catastrophic behavior driven by fear of injury might prolong the duration of acute pain, converting into chronic pain and causing significant disability and reduced quality of life [19,20]. Kinesiophobia may alter somatosensory changes affecting musculotendinous and capsule-ligamentous structures and affecting afferent proprioceptive input to the higher centers and impairing lumbar proprioception [21]. LJPS is estimated based on the ability of an individual to actively reposition the lumbar spine to a target position, and this reposition accuracy is measured in degrees [22,23]. Previous studies have shown a strong relationship between LJPS, kinesiophobia, pain intensity, and functional disability [24,25,26]. However, the evidence regarding how kinesiophobia impacts LJPS in CLBP is limited and has yet to be explored. To better understand and explain these relationships in patients with CLBP, synthesizing the evidence of the connection between kinesiophobia and LJPS would enable better clinical decision making and formulate effective treatment strategies.

The bio-psycho-social framework explains that functional impairment is brought on by a confluence of elements, including pain severity and bio-psychological issues [27]. In this clinical arena, the fear-avoidance model (FAM) explains the relationship between pain and disability and their contribution to developing chronic pain via the psychological process [28]. Fear of movement and catastrophizing thoughts further deteriorate the functional progression of the CLBP individual by increased disability and decreased quality of life [29]. Persistent pain often occurs among CLBP individuals [30]. Different authors have shown a significant relationship between pain severity, proprioceptive impairment, frequency of falls, and balance impairments [31]. Increased pain is associated with increased proprioceptive errors and decreased balance and functional mobility [32]. Previous studies have shown that pain is a significant factor that can increase fear of movement and impair motor control [33,34,35,36,37]. However, it is unknown how pain influences kinesiophobia and its relationship with LJPS in CLBP individuals. We employed mediation analysis using multiple regression to understand the relationship [27]. In line with the statement of the problem, the objectives of this study are (1) to evaluate the correlation between kinesiophobia and LJPS in CLBP subjects; (2) to compare LJPS between CLBP individuals and asymptomatic individuals; and (3) to assess if pain mediates the relationship between kinesiophobia and LJPS in CLBP individuals.

## 2. Materials and Methods

### 2.1. Study Design and Settings

This cross-sectional observational study was carried out in medical rehabilitation clinics at the Al-Qara campus, King Khalid University, Kingdom of Saudi Arabia, from May 2021 to December 2021. The local university ethics committee approved the study (ECM #2022-6012), and the study followed the Declaration of Helsinki principles.

### 2.2. Subjects

Eighty-three patients from the university hospital were referred to the physical therapy clinic after being diagnosed with non-specific CLBP following lumbar spine radiography and a complete screening process by an orthopedic doctor. A physical therapist, who was a blind evaluator and experienced in musculoskeletal assessments, evaluated kinesiophobia, LJPS, and postural control measures. Consecutive presentations of people with a referral for CLBP treatment to the physiotherapy department were screened. Participants who were clinically diagnosed with chronic LBP were recruited and further assessed for eligibility. The subjects were included in this study if they met the following criteria: (1) males and females aged between 20 and 60; (2) suffering from non-specific CLBP for more than 12 weeks; (3) having a pain intensity of 3 or higher on a 0 to 10 visual analog scale; and (4) willing to participate in the study. The subjects were excluded if they were the following: (1) subjects with specific back pain (fracture, osteoporosis or degenerative changes, prolapse intervertebral disc, bone disorders, arthritis, tumor); (2) subjects with neurological involvement (radiculopathy, myelopathy); (3) subjects with previous spinal surgery; (4) subjects with spinal infections; (5) subjects with a severe psychiatric disorder; and (6) subjects unable to understand or follow examiner commands. Asymptomatic subjects were included if they were over 18 years, healthy, and able to follow the commands of the examiner. If they were using pain medication or had a history of back pain during the previous six months, they were excluded. All the recruited individuals provided written consent before the commencement of the study.

### 2.3. Outcome Measures

#### 2.3.1. Pain Intensity

The subjects’ current level of CLBP intensity was assessed using a visual analog scale (VAS). The VAS is a 0 to 10 mm line that denotes “0” as no pain and “10” as the worst pain. The VAS is the most frequently used and reliable instrument in assessing the severity of pain in LBP individuals [38].

#### 2.3.2. Kinesiophobia

The Tampa Scale of Kinesiophobia (TSK) scale was first developed to assess fear of movement in individuals with CLBP [39]. It is a 17-item self-report questionnaire with a score range between 17 and 68. A score of 17 indicates the absence of kinesiophobia, and 68 represents the highest fear of movement [40]. A cut-off TSK score of ≥37 indicates fear of movement [41]. The TSK has presented a valid and reliable tool to indicate the presence of kinesiophobia in chronic non-specific CLBP individuals [42].

#### 2.3.3. Lumbar Joint Position Sense

LJPS was determined using a dual digital inclinometer (Dualer IQ—Midvale, UT, USA). The LJPS test, which assesses the ability to recognize and reproduce lumbar spine positions, frequently necessitates the time-consuming task of analyzing images using particular software and/or high-tech tools, such as an isokinetic dynamometer, inertial sensors, an electro goniometer, or photo analysis (high-resolution camera), to monitor proprioceptive deficits in individuals with CLBP [22,43]. These measurement tools are trustworthy and accurate, but they are not portable, and their installation takes time. In order to rapidly and efficiently obtain proprioceptive errors, some authors have created and tested digital inclinometers. On the other hand, when compared to pricey and sophisticated equipment, the digital inclinometer is an economical, easy-to-use gadget that takes up less space, is operated by a single rater, and takes measurements quickly. A digital inclinometer is a reliable instrument (r = 0.98) that allows physical therapists to accurately assess the range of motion of joint position errors [22,44]. LJPS errors were measured in degrees in directions of lumbar flexion, extension, and side-bending left and right as an estimate of LJPS.

All the tests were performed in a calm and well-ventilated lab. All the subjects were blindfolded during the LJPS testing procedure. We adopted the target lumbar positioning testing protocol by Reddy et al. [22]. The individual repositions his lumbar spine to a target position from the neutral position. To measure LJPS in lumbar flexion and extension, the primary inclinometer was positioned over the lateral aspect of the chest at the T12 level, and the secondary Inclinometer was placed over the hemipelvis at the S1 level (Figure 1).

To measure LJPS in side-bending left and right, the primary inclinometer was placed on the upper back (over the T12 spinous process) and the secondary inclinometer over the mid and central aspect of the sacrum [22]. All subjects’ full range of motion (ROM) was measured (flexion, extension, and side-bending left and right), and 50% of the available ROM was selected as their target to be repositioned during the repositioning task [22].

To start the testing, the subjects were asked to stand straight and were asked to determine their self-selected neutral spine position. The examiner gently guided the participants to the target position (50% of the available ROM), and they were maintained in this position for a period of five seconds and asked to memorize [22]. Following this, the examiner guided the individual back to the starting position. Next, the examiner asked the subjects to move actively and reposition their lumbar spine to the target position; once the participants reached the target position, they intimated the test by saying, “YES”, and the reposition errors were computed as LJPS in degrees [22]. LJPS was evaluated in lumbar flexion, extension, and side-bending left and right bending directions. Each test was repeated three times, and the mean value was used as the reposition accuracy value.

### 2.4. Sample Size

G*power statistical software estimated the sample size using the following formula: one study group vs. population and continuous variables [45]. A study group mean of 4.8, known population means of 3.6 (SD = 2.2), 1-β (statistical power) of 0.80, and an α (significance level) of 0.05 were used. The estimated sample was 80 in each group.

### 2.5. Statistical Analysis

The Shapiro–Wilk test was used to assess the normality of the study variables, and the data followed a normal distribution. The relationship between the TSK score (kinesiophobia) and LJPS was estimated using Pearson’s correlation coefficient. ANOVA was used to compare LJPS between the CLBP group and the asymptomatic group. Mediation analysis was computed to assess the impact of pain on the relationship between kinesiophobia and LJPS in CLBP individuals. The mediation analysis included a three-step process (Figure 2).

Bivariate regression assessed the total effect between TSK scores (kinesiophobia) and LJPS (step 1). The direct effect between pain and TSK score (pathway A) was assessed using bivariate regression (step 2). Multiple regression was used to assess the direct effect between TSK score and LJPS (Pathway C) and between TSK score and pain (Pathway B) as step 3. SPSS ver. 24 (SPSS Inc., Chicago, IL, USA) was used to conduct statistical analysis and the significance level was determined at *p* < 0.05.

## 3. Results

Eighty-three subjects with CLBP (mean age of 48.9 ± 7.5 years) and 105 asymptomatic individuals (mean age of 49.4 ± 7.0 years) participated in this study. The demographic and physical characteristics of the study subjects are displayed in Table 1. The individuals were overweight, as shown by BMI (26.4 ± 3.3 kg/m^2^). The CLBP individuals had a mean TSK score of 41.2 ± 3.2. LJPS error was larger in the CLBP group compared to the asymptomatic group. The proprioception was impaired significantly in the CLBP group (*p* < 0.001) in all the directions tested (Table 1).

Kinesiophobia showed significant positive correlations with LJPS, as summarized in Table 2.

TSK scores showed moderate correlations with JPS in flexion (r = 0.51; *p* < 0.001), extension (0.41; *p* < 0.001), and side-bending left (r = 0.37; *p* = 0.001), and side-bending right (r = 0.34; *p* = 0.002) directions.

The results of the mediation analysis are summarized in Table 3 and Table 4.

As shown in Figure 2 in this mediation model, the total effect was the observed effect of kinesiophobia on LJPS (pathway C). Kinesiophobia was significantly associated with LJPS (flexion: B = 0.22, *p* ≤ 0.001, extension: B = 0.23, *p* ≤ 0.001, side-bending left: B = 0.18, *p* = 0.001, side-bending right: B = 0.16, *p* = 0.002). The total effect also decomposed into the direct effect of kinesiophobia on LJPS (pathway C′) and the indirect effects of kinesiophobia on LJPS through pain (mediated: pathway A + B). The indirect effect was statistically significant (Sobel test), in which pain explained the association between kinesiophobia and LJPS (*p* > 0.05).

## 4. Discussion

The present study investigates the relationship between kinesiophobia and LJPS. Furthermore, it assesses if pain could mediate the relationship between Kinesiophobia and LJPS in CLBP individuals. The individuals in this study demonstrated significant correlations (moderately positive) between kinesiophobia and LJPS. Pain mediated the relationship between kinesiophobia and LJPS in CLBP individuals.

In this study, kinesiophobia significantly correlated with LJPS, indicating that fear of movement can be a coping strategy for impaired proprioceptive acuity. This relationship suggests that fear of movement significantly affects the lumbar motor control [46]. Increased fear of movement and catastrophic behavior in CLBP patients can reduce muscle strength, endurance, and functional capacity [47,48]. Previous studies have demonstrated a significant positive correlation between reduced muscle strength, endurance, and functional performance to impaired JPS [48,49,50]. This mechanism can explain the relationship between kinesiophobia and LJPS. Limited studies have evaluated the impact of kinesiophobia on LJPS in CLBP individuals. Similar to our study methods, Kandakurti et al. [26] assessed the impact of kinesiophobia on lumbar extensor endurance and position sense in patients with CLBP, and this study showed that individuals with increased TSK scores had decreased lumbar extensor endurance and high LJPS.

Our study results are in accordance with the study conducted by Alshahrani et al. [33], in which kinesiophobia had a moderate association with knee joint position sense (r = 0.38 to 0.5, *p* < 0.05) in knee osteoarthritis individuals. Furthermore, the associations were significant in different target joint position sense angles (15°, 30°, and 60° of knee flexion) tested (*p* < 0.05). This implies that agonists and antagonists contract rhythmically to contribute to effective motor control and force-generating capabilities. It also implies that kinesiophobia can influence these components and alter the afferent proprioceptive input leading to impaired joint position sense [51]. A study by Pakzad and colleagues [52] discovered that fear of movement changed muscle activation patterns and neuromuscular control during walking. This suggests that kinesiophobia is a factor that can disrupt proprioception because it changes motor control and has an adverse impact on afferent proprioceptive input, impairing JPS. A study by Asiri et al. [34] demonstrated that cervical proprioception is impacted by fear of movement and showed a moderate positive association between TSK scores and cervical joint position sense errors in extension and rotation directions (r range between 0.31 and 0.48, *p* < 0.05). Furthermore, in the study conducted by Asiri et al., kinesiophobia significantly predicted cervical proprioceptive acuity [34]. The results of these studies confirm that kinesiophobia significantly impairs JPS. Contrary to our study results, the study by Aydo˘gdu et al. [53] in subjects with ACL reconstruction did not show an association between fear of movement and knee joint position sense. The explanation for the difference is that the mean TSK score for kinesiophobia in our study is higher (41.2 3.2) than that found by Aydo˘gdu et al. (36.54 4.22). LJPS may have been affected by participants’ greater TSK levels and chronic back pain in this study.

Fear of movement for an extended period and increased pain intensity can lead to impaired lumbar proprioception [54]. A vicious loop exists of pain, muscle fatigue, atrophy, and impaired proprioception in subjects with CLBP [54,55]. Karayannis et al. [56] showed that increased pain intensity and chronicity, negative thoughts, and fear-avoidance behavior could significantly impact position sense and bodily balance [56].

We anticipated that pain might mediate between kinesiophobia and LJPS. With many musculoskeletal disorders, the fear of pain prolongs the acute pain course and helps it progress into a chronic condition [57]. In addition, proprioception may be compromised by decreased flexibility, deconditioning, and loss of muscle tone brought on by increased pain, chronicity, and disuse [58]. Pain has the ability to influence numerous aspects of the nervous system, including the sensitivity of muscle spindles and the way the central nervous system modulates proprioceptive afferent signals [59]. Asiri et al. [37] conducted a study investigating the mediation effect of pain on the relationship between kinesiophobia and postural control in fibromyalgia syndrome, and the results demonstrated that pain significantly mediated their relationship to produce altered motor control, hence impacting balancing ability. Similarly in our study, pain may have impacted the relationship between kinesiophobia and LJPS. Furthermore, fear of movement is significantly influenced by the presence of pain and is intensified by the chronicity of pain. Higher TSK scores (which indicate higher levels of kinesiophobia) were seen in individuals with more pain, as measured by their VAS scores. These individuals exhibit impaired motor control affecting LJPS. Although our investigation revealed the mediating influence of pain on kinesiophobia and LJPS, no studies have studied the pathophysiology behind this effect. Additional research is warranted to determine whether pain-relieving interventions can decrease the fear of movement and improve position sense in CLBP individuals.

### 4.1. Practical Clinical Implications

This study revealed that patients with CLBP had decreased proprioception compared to asymptomatic individuals, and earlier research has demonstrated that this population falls more frequently [31]. These results corroborate other studies that have suggested individuals with CLBP fell more frequently and have shown that kinesiophobia can make patients with CLBP more susceptible to balance issues [60,61]. For individuals with CLBP who are undergoing rehabilitation, the study’s findings have clinical ramifications. Moreover, kinesiophobia played a major role in impaired motor control associated with CLBP, and therapeutic approaches for patients with CLBP may take this into account.

### 4.2. Future Research Implications

We noticed that people with CLBP had higher TSK scores and more balance issues as measured by LJPS. A higher incidence of falls is directly correlated with decreased proprioception [62,63]. Future research should examine the direct connection between the number of falls and TSK scores. Additionally, evaluating degrees of kinesiophobia and their association with the frequency of falls across age groups and genders would yield crucial data for comprehending and treating individuals with CLBP.

Cognitive behavioral therapy (CBT) is a psychological approach that aims to eradicate negative feelings and behaviors, while also modifying patients’ thoughts, beliefs, and behaviors in order to correct poor cognition [64]. It is appropriate for people without mental illnesses and is characterized by integrity, initiative, enthusiasm, and a brief course of treatment [64]. According to studies, CBT not only helps patients with kinesiophobia by identifying and treating errors in automatic thought and poor cognitive behavior, but it also boosts patients’ self-efficacy, decreases anxiety and sadness, increases physical activity, and enhances quality of life [65,66]. Further research should be conducted to determine the effectiveness of CBT in lowering kinesiophobia in CLBP participants and how it affects LJPS.

### 4.3. Limitations of the Study

This study has a few limitations. Even though we tested LJPS with a digital inclinometer, a motion analysis system is the gold standard for measuring LJPS, but it is expensive, difficult to operate in a clinical environment, and cannot be carried out for testing on the field. On the other hand, when compared to expensive and sophisticated equipment, the digital inclinometer is an affordable, easy-to-use gadget that takes up less space, is operated by a single rater, and takes measurements quickly. This survey reflected the limited population of Saudi Arabia (recruited from a university clinic) suffering from CLBP. We did not evaluate the associations between functional outcome assessments and kinesiophobia. Future studies should assess these factors to understand their relationship and plan rehabilitation strategies for CLBP subjects.

## 5. Conclusions

This study demonstrated that individuals with CLBP had moderate kinesiophobia and were positively correlated with LJPS. LJPS is impaired in CLBP individuals compared to asymptomatic individuals. Nonetheless, the findings are expected to contribute to the scientific base of experts involved in the physical examination and rehabilitation of individuals with CLBP experiencing kinesiophobia. Furthermore, our results suggest that pain mediated the relationship between kinesiophobia and LJPS. This should be considered by therapists or clinicians when developing preventive and rehabilitative programs for CLBP patients. CBT may reduce kinesiophobia, and referral to a health counselor to help patients with CLBP and future studies should assess the efficacy of CBT in reducing kinesiophobia and its effect on LJPS.

## Figures and Tables

**Figure 1 ijerph-20-05193-f001:**
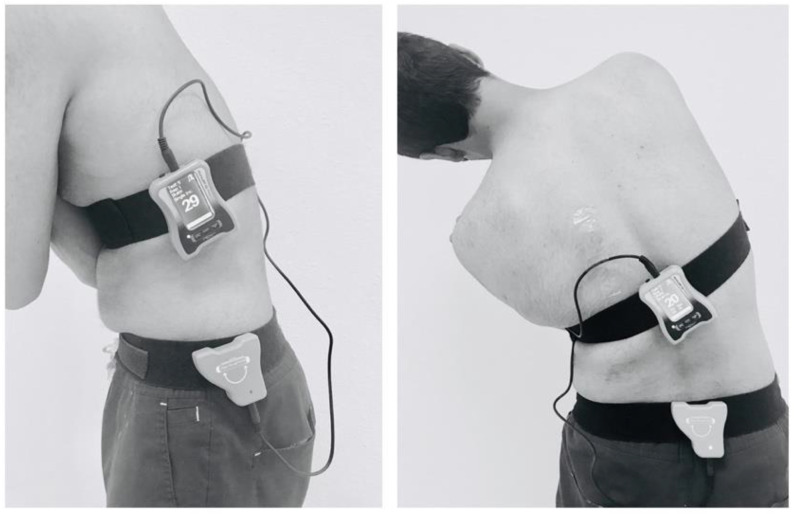
Assessment of lumbar joint position sense using a digital inclinometer.

**Figure 2 ijerph-20-05193-f002:**
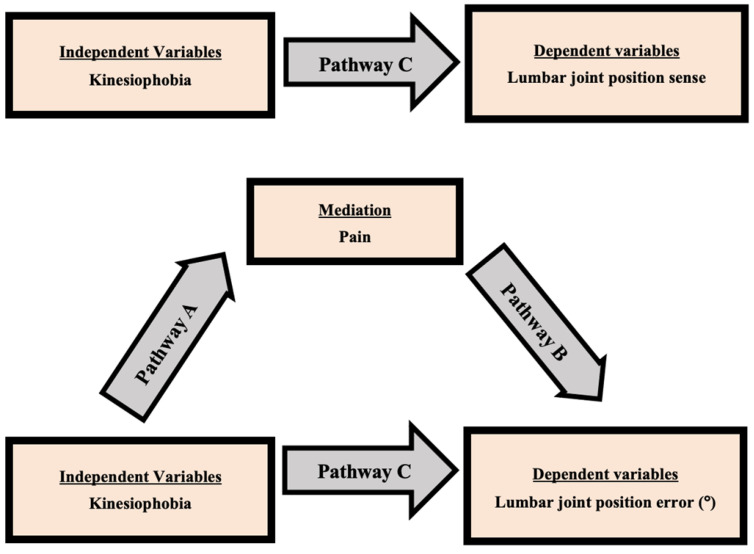
Model of the potential mediating effect of pain on the relationship between kinesiophobia and lumbar joint position sense.

**Table 1 ijerph-20-05193-t001:** Physical and demographic characteristics of participants.

Variables	CLBP Group (n = 83)	Asymptomatic (n = 95)	*p*-Value
Age (years)	48.9 ± 7.5	49.4 ± 7.0	0.686
Gender (Male/Female) (n%)	49:34	62:33	0.727
BMI (kg/m^2^)	26.3 ± 7.4	25.8 ± 8.6	0.198
Duration of back pain (Months)	22.3 ± 10.8	-	-
Use of medication (yes/No) (n%)	58 (69.9%)/25 (30.1%)	-	-
VAS (0–10 cm)	4.9 ± 1.8	-	-
ODI (0–100%)	31.7 ± 5.0	-	-
Kinesiophobia	41.2 ± 3.2	-	
LJPS in flexion (°)	4.9 ± 1.4	2.4 ± 1.2	<0.001
LJPS in Extension (°)	5.2 ± 1.3	1.4 ± 1.2	<0.001
LJPS in side-bending left (°)	4.8 ± 1.5	2.9 ± 0.7	<0.001
LJPS in side-bending right (°)	4.8 ± 1.5	2.8 ± 0.6	<0.001

BMI = body mass index; VAS = visual analog scale; ODI = Oswestry disability index; LJPS = lumbar joint position sense.

**Table 2 ijerph-20-05193-t002:** Coefficient of correlation between kinesiophobia and lumbar joint position sense in CLBP individuals (n = 83).

Test Variables	Kinesiophobia (TSK Score)	
	r	*p*-Value
LJPS in flexion (°)	0.51	<0.001
LJPS in extension (°)	0.41	<0.001
LJPS in side-bending left (°)	0.37	0.001
LJPS in side-bending right (°)	0.34	0.002

TSK = Tampa Scale for Kinesiophobia, LJPS = lumbar joint position sense.

**Table 3 ijerph-20-05193-t003:** Mediation analysis of the association between kinesiophobia and lumbar JPS.

Test Variables	Total Effect—Direct and Indirect			Direct Effect			Indirect Effect		
	B	SE	*p*-Value	B	SE	*p*-Value	B	SE	*p*-Value
Pain × LJPS in flexion (°) × TSK	0.22	0.04	<0.001	0.22	0.04	<0.001	0.06	0.01	0.001
Pain × LJPS in extension (°) × TSK	0.23	0.05	<0.001	0.21	0.03	<0.001	0.05	0.01	0.001
Pain × LJPS in side-bending left (°) × TSK	0.18	0.05	0.001	0.18	0.05	0.001	0.10	0.02	0.002
Pain × LJPS in side-bending right (°) × TSK	0.16	0.05	0.002	0.17	0.05	0.001	0.28	0.02	0.002

LJPS = lumbar joint position sense; TSK = Tampa Scale of Kinesiophobia; B = unstandardized coefficients; SE = standard error.

**Table 4 ijerph-20-05193-t004:** Sobel test for indirect effect for statistical significance.

Test Variables	Sobel-Test	SE	*p*-Value
Pain × LJPS in flexion (°) × TSK	0.29	0.03	0.030
Pain × LJPS in extension (°) × TSK	0.24	0.02	0.020
Pain × LJPS in side-bending left (°) × TSK	0.15	0.03	0.018
Pain × LJPS in side-bending right (°) × TSK	0.78	0.08	0.024

LJPS = lumbar joint position sense; TSK = Tampa Scale of Kinesiophobia; SE = standard error.

## Data Availability

All data generated or analyzed during this study are with the corresponding author. The data will be provided on request.

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
