# Peer review of "The Mediation Effect of Pain on the Relationship between Kinesiophobia and Lumbar Joint Position Sense in Chronic Low Back Pain Individuals: A Cross-Sectional Study"

_ijerph, 2023, doi:10.3390/ijerph20065193_

Round 1
Reviewer 1 Report (New Reviewer)
This is a nice done study. Compared individuals with CLBP the kinesiophobia and found positive correlation with lumbar joint position sense (LJPS) which is expected. Pain remediation strategies are discussed. Authors have discussed the limitations of this study as well.
Author Response
We thank you and the reviewers for your efforts and time in reviewing our manuscript.
Reviewer 2 Report (New Reviewer)
Thank you for your submission and corrections based on the reviewers. The comments have been attached.

Author Response
Please find the attached file

Reviewer 3 Report (New Reviewer)
Dear Author
This is a new paper exploring the causes of the association between CLBP and LJPS and is of high clinical significance.
On the other hand, there are a few things that need to be corrected, see below.
Major points
l The abstract does not mention lumbar extension. The results should be accurately described.
l Table 2 and Figure 3 show the same meaning. Consider deleting Figure 3.
Mainor points
・Lunber flexion is mentioned twice. Please confirm. Could it be an error in extension?
・Eighty-three subjects with CLBP (mean age of 48.9 ± 7.5 years) and 105 asymptomatic
(mean age of 49.4 ± 7.0 years) participated in this study with a.
→Delete 'with a'.
Author Response
Please find the attached file

Reviewer 4 Report (New Reviewer)
Research methodology needs reliable improvements.
Thank you for the opportunity to review this article
1. Abstract - the required structure is not maintained.
2. Introduction - requires ordering of posted information, is chaotic
3. What practical-life implications are to come from this study?
4. 2. Materials and Methods - The description of the study group is very superficial.
5. Verse 111 - Here the abbreviation LBP is used, previously CLBP was used.
6. Vesre 116 - The Tampa Scale of Kinesiophobia (TSK) - why did the authors use this tool and not another?
7. Verse 123 - digital inclinometer: why did the authors use equipment with a large error measurement? Why isn't there a detailed description of making a measurement with an inclinometer?
8. The methodology must be carefully designed.
9. 2.4. Sample size - Did the authors evaluate the power of the statistical test?
10. 2.5. Statistical Analysis - the authors did not state what level of significance they were referring to.
11. Verse 190 - "....was performed using SPSS version 24.0 software." - no reference to references.
12. Limitations - The authors are aware of the limitations of using an inclinometer and yet they use it for their research.
13. Conclusion - first of all, patients with kinesiophobia should receive therapy from psychologists and psychiatrists.
Author Response
Please find the attached file

This manuscript is a resubmission of an earlier submission. The following is a list of the peer review reports and author responses from that submission.
Round 1
Reviewer 1 Report
Thank you very much for conducting this interesting piece of research. While I believe that this should be made available to readers, I do have some concerns which manly refer to the interpretation and discussion of the correlation/regression analyses results. Please find my detailed comments below:
Methods
· Was the study registered, this is not mentioned in the methods section
· Why the upper and lower age limits?
· Can you comment on why you did not measure lumbar extension and rotation joint position sense?
· In the methods section, you only describe the patient group, can you please state where and how and according to which criteria you selected the control group?
· Statistical analysis should be ANOVA (instead of ANONA)
Results
· Table 2: not sure that explanatory variable is the right term. Isn’t this the test variable that was correlated with TSK?
· Can you explain the plots in Figure 3? They are left somewhat uncommented.
Discussion
This needs to be entirely rewritten, since the way the discussion line is presented seems to imply that there is a causal relationship between joint position sense and kinesiophobia. The analyses rely on correlations and regression models. Both are not designed for causal relationships. Pain mediated the relationship between TSK and JPSE, hence, it is possible that a person in pain has kinesiophobia (because it hurts to move) and has an impaired JPS (because feedback from painful structures is not precise, maybe based on somatosensory or motor cortex changes) but these two are causally totally unrelated.
Next possibility: it is not the kinesiophobia that influences joint position sense but vice versa. Can you exclude that not being able to precisely feel your body in space makes you more fearful of moving normally (because you might end up in a position that hurts without wanting it?).
Especially when having to rely on correlations, it is extremely important that all implications are discussed.
· line 212: “dependent on drugs” – please rephrase,
· lines 213/214 – these are very general statements. Please be more precise and use references to support your statements. Is this really an outcome of your study or rather general features of low back pain (and should therefore be moved to the introduction)
· lines 241ff only listing other references is not a discussion
Minor language issues, mainly missing or additional spaces ….some tempus errors, e.g. who “is” a blinded evaluator should be “was” or multistep regression “will be” used should be “was” – kinesiophobia in the discussion spelled with a capital “k”.
Reviewer 2 Report
-The authors In the introduction section, has not explained well the importance of the relationship between Kinesiophobia and the sense of back position. They should explain more about the relationship between the pain- matrix and kinesiophobia and then its effect on the joint position sense .
-The novelty of work is not determined with considering the publish article :Infuence of kinesiophobia on pain intensity,
disability, muscle endurance, and position sense in patients with chronic low back pain—a case-control study by praveen Kumar Kandakurti1, Watson Arulsingh and Sharad S Patil.
this reference was not in references list.
-In the research method, the need to express more details about
The number of trials and The intensity of the pain which is not be clarified, related to the day of the test or the experience of the subject on in his / her lifetime
-There are some edit problems in the text of errors, for example, in line 62, it is not necessary to mention the word error. This word is mentioned by JPE word.
-In the method part from line 118-132 is needed reference.
-In the method part and explanation of sample size, should be rewritten .the explanation for this part not professional and standard manner.
Discussion are very poor . More of them is about other joints such as cervical spine discussed and knee ( from line218-234),also the discussion in related results of Pain as mediator in subjects with low back pain is poor and Insufficient.
The references 37 and 39 are not related to the written text.